# Influence of Different Dehydration Levels on Volatile Profiles, Phenolic Contents and Skin Hardness of Alkaline Pre-Treated Grapes *cv* Muscat of Alexandria (*Vitis vinifera* L.)

**DOI:** 10.3390/foods9050666

**Published:** 2020-05-21

**Authors:** Onofrio Corona, Diego Planeta, Paola Bambina, Simone Giacosa, Maria Alessandra Paissoni, Margherita Squadrito, Fabrizio Torchio, Susana Río Segade, Luciano Cinquanta, Vincenzo Gerbi, Luca Rolle

**Affiliations:** 1Dipartimento di Scienze Agrarie, Alimentari e Forestali, Università degli Studi di Palermo, 90128 Palermo, Italy; diego.planeta@unipa.it (D.P.); paola.bambina@unipa.it (P.B.); margherita.squadrito@unipa.it (M.S.); luciano.cinquanta@unipa.it (L.C.); 2Dipartimento di Scienze Agrarie, Forestali e Alimentari, Università degli Studi di Torino, 10095 Grugliasco, Italy; simone.giacosa@unito.it (S.G.); mariaalessandra.paissoni@unito.it (M.A.P.); fabrizio.torchio81@gmail.com (F.T.); susana.riosegade@unito.it (S.R.S.); vincenzo.gerbi@unito.it (V.G.); luca.rolle@unito.it (L.R.)

**Keywords:** Zibibbo, postharvest dehydration, alkaline pre-treatments, aroma compounds, linalool, *Passito* wine

## Abstract

A dehydration experiment was carried out on *Vitis vinifera* L. *cv* Muscat of Alexandria (synonym Zibibbo) following the process for the production of renowned special dessert wines produced on Pantelleria island (Sicily, Italy). Harvested berries were pre-treated in a sodium hydroxide dipping solution (45 g/L, dipped for 185 s, 25 °C) to accelerate the drying process, rinsed, and dehydrated in simulated conditions (relative humidity 30%, 30 °C temperature, air speed 0.9 m/s). Three dehydration levels were achieved, corresponding to “*Passolata*”, “*Bionda*”, and “*Malaga*” stages (35%, 50%, and 65% of weight loss, respectively) of the Pantelleria denomination of origin (DOC). Grape skin mechanical properties, technological parameters, phenolics, and aroma profile varied considerably during dehydration. The most important aroma compounds for their olfactory impact, such as linalool, geraniol, nerol, and citronellol, especially in glycosylated forms, significantly increased in dried grapes compared to fresh ones, even if aroma profile modification occurred. A decrease in break skin force could have induced higher release of flavonoids. The findings showed relevant changes, allowing winemakers to better select the ratio of fresh and dehydrated grapes in the function of the final desired wine.

## 1. Introduction

Wines produced from overripe and partially dehydrated grapes are a centuries-old tradition. Many historical special wines, such as Icewines, Sauternes and German wines (Botrytized wines), Porto, Marsala, Tokaj, Pedro Ximénez, Vin de Paille, and Amarone, are obtained in several winegrowing areas of the world from both red and white grapes, using different production technologies [1]. In the last few years, consumer demand for diversified oenological products led to an increased production and research interest in special wines, including *Passito*, Fortified, and Reinforced wines produced by grape dehydration or withering [1].

Winegrape withering is a dynamic process of water loss whereby berries are partially dehydrated at different levels under controlled or uncontrolled environmental conditions [2,3,4]. Winegrape’s postharvest dehydration is accomplished by off-vine dehydration through direct sun exposure in favourable conditions [5,6,7], indoors in naturally ventilated rooms called “*fruttai*” [8], or in chambers with thermohygrometric and airflow control [9]. The main result of water loss is an increased potential concentration of sugars and several metabolites in the pulp and skins, although modifications on the aromatic and phenolic characteristics occur and may vary in their function based on the dehydration techniques, environmental conditions, and varietal characteristics. Indeed, beyond the postharvest water loss effects on primary metabolism (shift from aerobic to anaerobic respiration), the secondary metabolism (particularly volatile compounds and polyphenols) also changes to different extents depending on the rate and amount of water loss [10,11,12,13,14]. With regards to secondary metabolites, a first metabolic stress response occurs, involving changes in membrane permeability by activation of lipoxygenase (LOX) and alcohol dehydrogenase (ADH) enzymes, of which promote the formation of different volatile compounds [10,11].

Three environmental factors are the main drivers of the water loss trend: temperature, relative humidity, and airflow. Nevertheless, varietal differences and ripeness at harvest characteristics can influence dehydration kinetics. Among them, berry skin hardness of harvested berries, which is related to variety and seasonal climatic condition [15], may influence the dehydration rate. Particularly, grape dehydration rate significantly decreases with increasing skin hardness [16,17]. Therefore, knowledge of the variety of features together with the choice and management of dehydration conditions is of fundamental importance in producing the desired final products.

Among the technologies applied to grape dehydration, chemical pre-treatments have been proposed in order to control water loss speed, aiming to decrease the time required to obtain the correct dehydration level. Chemical pre-treatments are widely used in raisins production to increase the dehydration rate through microstructural changes in the epicuticular wax layer covering the grape cuticle, which enhance the permeability of the grape skin and facilitate moisture diffusion [18,19,20]. These pre-treatments consist of dipping the berries into alkaline emulsions of ethyl or methyl esters, sodium hydroxide, or potassium carbonate for several minutes [7,19,21], leading to skin breakage and overall tissue softening. However, concentration and the nature of the chemical agent, dipping time, and temperature are the most important factors influencing the different extent of the texture degradation, which is related to enzymatic and non-enzymatic changes in the cell wall structure [22]. These modifications were particularly evident with sodium hydroxide solutions, given by its ability to solubilize large amounts of both pectic substances and xyloglucans [23].

In the production of Sicilian special wine belonging to Pantelleria Controlled Denomination of Origin (DOC), three types of dried grape of the cultivar Muscat of Alexandria (synonym Zibibbo), pre-treated with alkaline solutions, are used. These are called “*Passolata*”, “*Bionda*”, and “*Malaga*” and they are characterized by different levels of dehydration, i.e., 35%, 50%, and 65% weight loss, respectively. More specifically, different types are allowed depending on the winemaking process and on the type of dehydrated grapes used. Briefly, for the production of “*Pantelleria DOC-Moscato dorato*” and “*Pantelleria DOC-Moscato liquoroso*”, musts obtained from fresh and/or overripe grapes are added with different percentages of “*Passolata*” grapes. Instead, for the production of “*Pantelleria DOC-Passito*” and “*Pantelleria DOC-Passito Liquoroso*”, the grapes “*Bionda*”, and “*Malaga*” are used. In this case, the alcoholic fermentation of the base wine is blocked at 8%–10% (v/v) of ethanol, generally by refrigeration at low temperatures, and in this step, the dehydrated grapes are added as whole berries for a new fermentation in order to sweeten the final product [2]. In addition, in the “*Pantelleria DOC-Passito Liquoroso*”, the alcohol deriving from winemaking processes is allowed for fortification purposes [1].

Aroma compounds are particularly relevant in *Passito* wines and are strongly influenced by vineyard practices, *noble rot* presence, grape characteristics, and dehydration conditions [3,4,24,25,26,27], as well as winemaking technology and evolution [2,28,29]. In particular, using *cv* Muscat of Alexandria, the relationship between grape volatile content and wine quality is dependent on the terpene content and profile. In fact, the terpenoid compounds are closely associated with the sensory expression of this wine bouquet, contributing to flowery odours, which are used for variety characterization [2,28,30,31]. The monoterpenes can be present in grapes as both free and glycosidically conjugated forms, with the free volatile aroma contributing to the olfactory impact of the derived wines. Their glycoside forms are quantitatively the most important and although they do not have a direct contribution to wine aroma, they represent the grape aromatic potential, since they are hydrolysed to free volatile during fermentation by yeast and by the acidic condition [25,32,33].

In this study, the evolution of the dehydration process of *cv* Muscat of Alexandria (*Vitis vinifera* L.) grape cultivar treated by alkaline solutions has been followed in order to simulate a typical dehydration process for the production of Pantelleria DOC wines. Fresh and dehydrated grapes corresponding to “*Passolata*”, “*Bionda*”, and “*Malaga*” dehydration levels were evaluated with a mean of technological, aromatic, and polyphenolic parameters. In particular, free and conjugates aroma compounds were analysed during dehydration and their ratio of degradation/concentration was discussed in order to better understand the physiological and technological aspects connected to the peculiar process for this wine production. Moreover, since the winemaking practice involves skin contact, phenolic compounds and skin mechanical properties were also investigated in order to understand phenolic extractability.

## 2. Materials and Methods

### 2.1. Grape Sampling, Alkaline Pre-Treatment, and Dehydration Process

Grape sampling, preparation, and treatments were done according to [7]. About 15 kg of *Vitis vinifera* L. *cv* Muscat of Alexandria grapes were harvested at technological ripeness (i.e., 22°Brix) from an experimental vineyard located in Mazara del Vallo (Sicily region, southern Italy). Samples were collected in small clusters from the vineyard and grape berries were manually separated from the stalk maintaining attached short pedicels. Healthy and not damaged berries were selected and then sorted according to their density by flotation in sodium chloride solutions ranging from 100 to 190 g/L, corresponding to densities ranging from 1069 to 1125 kg/m^3^ [7]. In order to minimize the possible heterogeneity of berries ripening, only the three most representative classes were considered for the study, corresponding to densities of 1081, 1088, and 1094 kg/m^3^, which accounted for relative weights of 55, 25, and 20% w/w, respectively. The berries belonging to these three classes were rinsed with water and randomly selected to form groups of 30 berries, prepared by keeping into account the distribution (by weight) of the density classes selected. Each berry group was then introduced in a single layer into metallic supports with meshes of 0.8 × 0.8 cm^2^, a solution that is able to allow the correct pre-treatment and aeration condition.

The alkaline treatment was performed as follows: briefly, the berries were dipped for 185 s into a 45 g/L sodium hydroxide solution at 25 °C temperature, washed for 30 s with water at 25 °C, and then air-dried. Afterwards, the post-treatment dehydration was performed in a thermo-hygrometrically controlled chamber set at a temperature of 30 °C, relative humidity (RH) of 30%, and using an air speed of 0.9 m/s, with the aim to artificially simulate the conditions used for the dehydration of Muscat of Alexandria grapes in Sicily. For each set, the berries’ weight was measured during the process using a technical balance (Gibertini E1700, Modena, Italy) and the weight loss (WL) percentage was calculated as:WL [%] = 100 − (Wd/Wf × 100),
where WL [%] is the weight loss percentage, Wd is the weight of dehydrated samples, and Wf is the weight of fresh samples [17].

Samples were collected at ripening (fresh berries), at 35% WL (day 5), 50% WL (day 8), and 65% WL (day 13) to achieve the dehydration stages locally referred to as “*Passolata*”, “*Bionda*”, and “*Malaga*”, respectively.

### 2.2. Technological Parameters

Three replicates of 30 berries for each sample (fresh grapes and dehydration levels) were used to determine the standard physicochemical parameters. The grape juice was obtained by manual crushing and centrifugation at 4000× *g* rpm and 15 °C temperature for 15 min (Hettich 32R, Kirchlengern, Germany). Juice titratable acidity, expressed as g/L of tartaric acid, was determined by OIV-MA-AS313-01 method [34], while pH was evaluated by potentiometry using an InoLab 730 pHmeter (WTW, Weilheim, DE) [34]. Reducing sugars (glucose and fructose) were quantified by HPLC (Agilent Technologies 1200 series, Santa Clara, CA, USA) equipped with a refractive index detector. The analyses were performed isocratically at 0.8 mL/min and 65 °C with a 300 × 7.8 mm i.d. cation exchange column (Aminex HPX-87H) and a Cation H+ Microguard cartridge (Bio-Rad Laboratories, Hercules, CA, USA), using 0.0013 mol/L H_2_SO_4_ as the mobile phase [2].

### 2.3. Determination of Grape Volatile Composition

Three replicates of 30 berries for fresh grapes and for each of the three dehydration levels studied were weighed and processed following the procedure proposed by [35] and summarized by [36]. The berries were de-seeded and the pulp was separated from the skin with the addition of Na_2_S_2_O_5_ (100 mg). The skins were treated with 20 mL of methanol for 1 h to release aroma compounds and to inactivate glycosidase enzymes and then crushed with a laboratory blender by a high-speed Ultra-Turrax T25 (IKA Labortechnik, Staufen, Germany). The pulps were crushed separately with a laboratory blender by a high-speed Ultra-Turrax T25 (IKA Labortechnik, Staufen, Germany) and then mixed with the skins’ mixture. The skin and pulp mixture was centrifuged twice (7000× *g*, 15 min, 4 °C) and the solid residue was washed with tartaric acid buffer (pH 3.2). The final extract (250 mL) was then clarified with a pectolytic enzyme (0.1 g) without secondary glycosidase activity (Rapidase X-Press, DSM, The Netherlands) at room temperature for 2 h. 1-Heptanol was added as an internal standard (0.2 mL of 30 mg/L solution in 10% ethanol) to the samples. Afterwards, an aliquot was loaded onto a 5 g C18 reversed-phase solid-phase extraction (SPE) cartridge (Isolute, SPE Columns, Uppsala, Sweden), previously activated with 20 mL of methanol and then 50 mL of deionized water using a flow-rate of ca. 3 mL/min. and then rinsed with 100 mL of deionized water to eliminate sugars, acids, and other low molecular weight polar compounds. The free aromatic fraction was then eluted with 25 mL of dichloromethane. The eluate was dried over anhydrous Na_2_SO_4_ and concentrated to about 0.2 mL under a stream of nitrogen. This extract, containing free volatile compounds, was immediately analyzed by gas chromatography/mass spectrometry (GC/MS). Afterwards, the glycoconjugates aromas were finally eluted from the cartridge with 20 mL of methanol and the eluate was concentrated to dryness using a vacuum rotary evaporator set at 30 °C (Buchi R-210, Switzerland). This dried glycosides extract was dissolved in 5 mL of citrate-phosphate buffer (0.2 M, pH 5) and submitted to enzymatic hydrolysis with 50 mg of an AR-2000 commercial preparation with glycosidase side activities (DSM Oenology, The Netherlands) performed at 40 °C for 24 h. After 24 h, 0.2 mL of 1-heptanol (30 mg/L solution in 10% ethanol) was added as an internal standard and the volatiles generated by the enzymatic hydrolysis of glycosylated precursors were then extracted following the SPE method previously described. The dichloromethane extract obtained was dried over anhydrous Na_2_SO_4_, concentrated to 0.2 mL and kept at −20 °C until analysis. GC/MS analysis was performed with a Agilent 6890 Series GC system and Agilent 5973 Net Work Mass Selective Detector (Agilent Technologies) equipped with a DB-WAX column (30 m, 0.250 mm i.d., film thickness 0.25 μm; Agilent Technologies).

The GC-MS conditions used were reported by [37]. The detection was carried out by electron impact mass spectrometry in total ion current (TIC) mode using an ionization energy of 70 eV. The mass acquisition range was m/z 30–330. Volatile organic compounds were identified by comparison of the mass spectra and GC retention times with those of the pure commercial standard compounds or others prepared in our laboratory and by comparing their mass spectra with those within the NIST/EPA/NIH Mass Spectral Library database (Version 2.0d, build 2005). The concentration (µg/kg berries and µg/100 berries) of volatile compounds was determined as 1-heptanol equivalents.

### 2.4. Determination of Polyphenols Content

Three replicates of 10 berries of fresh grapes and of grapes for each level of dehydration were taken for phenolic compounds extraction and determination. Berry skins and seeds were manually removed from the pulp using a laboratory spatula and dried with absorbent paper. The berry skins were quickly immersed in 25 mL of a hydroalcoholic buffer at pH 3.2, containing 5 g/L tartaric acid, 2 g/L Na_2_S_2_O_5_, and 12% *v/v* of ethanol. The pulp was collected in a beaker containing Na_2_S_2_O_5_ (50 mg). Afterwards, the skins and pulp were separately homogenized with an Ultraturrax T25 (IKA Labortechnik, Staufen, Germany) and centrifuged in a PK 131 centrifuge (ALC International, MI, Italy) for 5 min at 3000 × *g* at 20 °C. The supernatant (berry skin extract and berry juice from pulp) was then used for analysis. In the case of the berry seeds, they were macerated for 7 days at 25 °C in the above-mentioned hydroalcoholic buffer solution and the extract was used for analysis. Total flavonoids index (TFI) was determined in berry skin extract, seed extract, and juice using a UV-1800 spectrophotometer (Shimazdu Scientific Instruments Inc., Columbia, MD, USA) after dilution with an ethanol:water:hydrochloric acid-37% 70:30:1 (v/v) solution [8,38].

### 2.5. Determination of Berry Skin Hardness by Texture Analysis

Skin hardness was evaluated using a non-destructive puncture test [39]. For each sample, a set of 30 berries was analysed using a Stable Micro Systems TA.XTplus equipment (Godalming, Surrey, UK) equipped with a 5 kg load cell, a HDP/90 perforated platform, and a P/2N needle probe. Test speed was set at 1 mm/s and the force-distance curve was processed using the Texture Exponent software (Stable Micro Systems) to determine the berry skin break force (Fsk, N) mechanical parameter.

### 2.6. Statistical Analysis

Statistical analyses were performed using the statistical software package SPSS (version 17.0; SPSS Inc., Chicago, IL, USA). The Tukey-b test for P < 0.05 was used to establish statistical differences by one-way analysis of variance (ANOVA).

## 3. Results and Discussion

### 3.1. Dehydration Kinetics and Berries Chemical Composition

Usually, a dehydration process conducted on berries pre-treated with an alkaline treatment leads to faster weight loss when compared with the normal on-wine or off-vine grape withering process [10,13,40,41,42]. Berries weight loss (WL) during dehydration, mainly due to water loss, was rapid during the first 8 days of dehydration (up to 50% WL, about 6% of WL for day) and then slowed down from day 8 to 13 (WL interval of 15%, about 3% of WL for day; Table 1). The absolute berry weight decreased consistently from 4.97 g for fresh berries to 1.63 g for the maximum dehydration tested (up to 65% WL). The decreased speed of water loss found during the process is in agreement with the berries lowered water content [43]. Indeed, drying kinetics are reported as not constant during the process because after certain dehydration values, the channels clogging limits the amount of water spread and evaporation into the atmosphere [43,44,45,46]. In particular, the dehydration phenomenon is explained by mass transfer, which is mainly dependent on water and sugar content. Water can spread in a liquid state by capillary action from an area of higher water content to another of lower content. It passes from the pulp to the skin as the speed of dehydration of the skin is greater than that of the pulp. The sugars retrieve the water via osmosis and at the same time, they partly move, with the water, as far as the cells of the skin surface that act as a barrier [43]. Here, the sugars retain a certain quantity of water that, as it is remaining in the matrix, does not reach the atmosphere (osmotic effect). A further phenomenon is that hygroscopicity counters the action of osmosis. With the increase in temperature, there is an increase in the requirement of water in the atmosphere and the physical state of the sugars also changes [45]. This phenomenon happens even at temperatures of almost 40 °C, similar to those reached during sun drying of Zibibbo grapes in Sicily island, from which the wine *Passito di Pantelleria* is produced [2].

The temperature at which the grapes were maintained in this study (30 °C ± 1 and 30% RH) would have prompted a considerably lower speed of dehydration had the grapes not been treated with sodium hydroxide. It is hypothesized that in an environment with basic pH, it is possible to remove part of the bloom of the berry and increase dehydration speed. As well as a change in relative humidity and the atmospheric requirement for water, the speed of diffusion would also change, especially if drying does not take place in a controlled environment. This phenomenon is also accentuated by the formation or exposure of micro-fissures on the berry surface, through which evaporation is facilitated. Other treatments (such as ethyl oleate) should also allow one to achieve such objectives. To confirm this, previous studies examined the effect of alkaline solutions, obtained with potassium carbonate (K_2_CO_3_), on the speed of dehydration [21,44,47,48,49]. In our conditions and according to previous findings, the greatest water loss recorded in the first 8 days seems to be due to this phenomenon (creation of preferential hydrophilic pathways with a quicker passage of water). During the successive stages of dehydration, from 8 to 13 days, lower WL percentages were found and the wrinkling of the skin was observed. This effect could be ascribed to pores clogging and to micro-fissures created on the skin by pre-treatment with sodium hydroxide, limiting the movement of water by the creation of a hydrophobic barrier.

Considering sugar contents (Table 1), fastest water loss corresponded to higher quantities of sugar accumulated in the outer part of the pulp. In our experimental conditions, sugar levels increased considerably from 189 g/L of fresh grapes to 266 g/L at 35% WL and 344 g/L at 50% WL (about 41% and 82% increase, respectively, compared with fresh grapes, *p* < 0.001). Reducing sugars rose less in the final phase (387 g/L at 65% WL, about a 105% increase compared with fresh grapes), in accordance with the lower water loss. The titratable acidity show significant changes during water loss and the decrease in total acidity values in the first stage of the dehydration process could be ascribed to the metabolism effect of malic acid, which is in agreement with a previous study (7). The pH value was not significantly affected by dehydration (*p* > 0.05) and ranged from 3.20 (fresh grape) up to 3.26 at 35% WL, whereas it barely changed between 50% WL and 65% WL (pH = 3.25 and 3.24, respectively [7]).

### 3.2. Free Volatile Compounds

In the present study, 19 free volatile compounds were identified and quantified in fresh and dehydrated grapes *cv* Muscat of Alexandria. These included 17 terpene compounds and two alcohols. In order to evaluate the evolution of volatile compounds both from a physiological (biosynthesis/degradation ratio) and technological point of view (final grapes concentration), the data are reported in μg/100 berries (Table 2) and μg/kg of berries (Table 3), respectively.

Considering the physiological aspects, as reported in Table 2, several free terpene monohydroxylate alcohols were affected by the different dehydration levels. Among these compounds, known as varietal markers with aromatic character of sweet, rose-like, flowery notes [28,29,30,33,50], linalool, and geraniol, are those quantitatively most important (532 and 269 µg/100 berries, respectively), which is in agreement with earlier studies on *cv* Muscat of Alexandria [5,35,51,52]. In contrast with these studies, lower values of nerol were found (36 µg/100 berries). Other terpene alcohols were found in relevant concentrations, such as 2,6-dimethyl-3,7-octadiene-2,6-diol and 3,7-dimethyl-1,7-octadiene-3,6-diol (391 and 94 µg/100 berries, respectively), *trans-* and *cis*-pyran-linalool oxides (192 and 94 µg/100 berries, respectively), and *trans*-geranic acid (217 µg/100 berries). Lower values of α-terpineol, both 8-hydroxy-linalool isomers, hydroxy-geraniol, *trans-* and *cis*-furan-linalool oxides, and hotrienol were found in fresh grapes, ranging between 15 and 35 µg/100 berries. Relevantly, hotrienol may be derived from 2,6-dimethyl-3,7-octadiene-2,6-diol as extraction artefact or by H^+^ catalyzed hydrolysis of 2,6-dimethyl-3,7-Octadiene-2,6-diol [53,54]. Small amounts of citronellol and 2,6-dimethyl-7-octadiene-2,6-diol (3 and 5 µg/100 berries, respectively) were also found.

Several significant differences were found during the dehydration process for terpene compounds, except the two furan-linalool oxide isomers, hotrienol, citronellol, 2,6-dimethyl-3,7-octadiene-2,6-diol, *cis*-8-hydroxy-linalool, and hydroxy-geraniol. After 5 days of dehydration, at 35% WL, the content of linalool dropped significantly (from 532 to 53 µg/100 berries, *p <* 0.001), whereas geraniol remained almost unchanged (from 268 to 215 µg/100 berries; *p >* 0.05). Subsequently, at 50% and 65% WL, the linalool content remained unchanged (42 and 51 µg/100 berries, respectively; *p >* 0.05), in contrast to geraniol, which dropped significantly (80 and 28 µg/100 berries, for 50% and 65% WL, respectively). At 35% WL, the concentration of *trans*-pyran linalool oxide also decreased (from 192 to 100 µg/100 berries, *p <* 0.001), as well as at 50% and 60% WL (44 and 24 µg/100 berries, respectively, *p <* 0.001).

Among the other terpenols, the content of *cis*-pyran-linalool oxide was reduced at 65% WL (49 µg/100 berries, *p <* 0.001). This increase was also found for 2,6-dimethyl-7-octadiene-2,6-diol (ranging from 5.2 to 26 µg/100 berries between fresh and 65% WL, respectively). These compounds’ increase may be explained by linalool decrease. In fact, linalool has been proposed as the substrate for conversion to higher oxidation state compounds such as hydroxy-linalool derivatives (2,6-dimethyl-3,7-octadiene-2,6-diol and 2,6-dimethyl-1,7-octadiene-2,6-diol) [35]. Particularly, the increases of these two diols during sun drying of Muscat of Alexandria grapes in Pantelleria island was previously reported by [35]. On the other hand, the significant increase of content of α-terpineol during the entire dehydration process, even if small, could be due to H^+^ catalysed reaction on linalool, nerol, and geraniol [55], which are by contrast reduced in our experimental condition. The same trend was found for *trans*-geranic acid, which progressively decreased from 217 µg/100 berries in fresh grapes to 19 µg/100 berries in 65% WL dehydration point.

Concerning the benzenoids found, the evolution of 2-phenylethanol did not show a regular trend because of the possible interference of yeasts in the production of this compound, due to micro-fermentation which may occur during the process [56], while the trend of benzyl alcohol showed an inconstant behaviour during dehydration (Table 2).

Free volatile compounds expressed in µg/kg of berries showed a relevant effect of the concentration given by the water loss by increasing volatile aroma compounds during the dehydration process (Table 3). In fact, reporting the data in μg/kg of berries is conditioned by the fact that the number of berries needed to form 1 kg of grapes increases with increasing dehydration and this aspect allows us to better represent the actual winemaking condition. Considering terpenes, data expressed in μg/kg showed a significant decrease in linalool, *trans*-pyran linalool oxide, and geranic acid during dehydration (all *p <* 0.001), which is in accordance with the reduction found when expressed as μg/100 berries (Table 2). By contrast, the contents of *trans*-furan linalool oxide, hotrienol, α-terpineol, 2,6-dimethyl-3,7-octadiene-2,6-diol, and 2,6-dimethyl-7-octadiene-2,6-diol increased (all *p <* 0.001). Considering other terpenes detected, an uneven trend was reported. In particular, the μg/kg contents of nerol and geraniol showed a trend increase at 35% WL (*p <* 0.05 for nerol; *p >* 0.05 for geraniol) and then a reduction at 65% WL. The opposite behaviour was found for linalool (the most abundant terpene alcohol found and typical aroma marker), with an initial strong decrease during the dehydration up to 35% WL that remained almost unchanged at 50% WL and then an increase at 65% WL, while the concentration of *cis*-piran linalool oxide increased at 50 WL% and decreased at 65% WL. This behaviour is in accordance with previous data found on *cv* Muscat of Alexandria drying in both sun-drying and controlled-room conditions [35].

The geraniol trend showed that this compound was slightly involved in the degradation reactions occurring just after NaOH pre-treatment, but not in the subsequent reactions during prolonged dehydration. On the other hand, linalool was very sensitive, leading to an initial drop of concentration in both 35% and 50% WL samples. The behaviour of these two alcohols suggested that radical oxidation may occur during the first stage of dehydration, speeded up by reactive oxygen species (ROS) [57]. Consequently, linalool, which is a tertiary alcohol, would be more sensitive to these reactions than geraniol, justifying its more rapid decrease. A second mechanism of terpene alcohols degradation may be imputable to catalysed H^+^ hydrolytic reactions [57], which could be responsible for the geraniol decrease. The hydrolysis is probably responsible for the increases of α-terpineol and hotrienol, while the ROS catalysed reactions may be involved in the increase of 2,6-dimethyl-3,7-octadiene-2,6-diol during dehydration (all *p <* 0.001).

Notably, considering the μg/kg berries data, it is possible to understand the final concentration of odorant compounds with respect to their sensory threshold. At the end of the dehydration (65% WL), linalool, geraniol, and hotrienol were found to be above their odour threshold (expressed as μg/kg berries) and they are among the mainly volatile varietal markers of *cv* Muscat of Alexandria [4,58].

In general, total content of terpenes expressed in µg/100 berries (Table 2) underwent a reduction of -28.0%, -47.1%, and -52.9% for 35, 50, and 65% WL, respectively, with respect to the fresh grapes (*p <* 0.01). However, the data expressed in µg/kg of berries (Table 3) showed an increased concentration during the dehydration process (+18.0%, +25.4% and +56.5% for 35%, 50%, and 65% WL respectively, with respect to fresh grapes, *p <* 0.05). Therefore, the aroma compounds concentration effect is higher than degradation reactions, although final grapes’ aromatic profile is changed depending on the susceptibility of individual compounds to several oxidative and hydrolytic reactions and on their reaction products accumulation. As well, the effects of pre-treatment and dehydration had an impact on the terpenes, considerably changing the aroma profile of dehydrated grapes compared to the fresh ones. The results obtained are in agreement with [4], whereas a slight contrast was found with the results of *cv* Muscat of Alexandria drying at different levels of dehydration reported by [5], demonstrating the complexity of subsequent reactions that are involved in water loss process. However, a shared result is that if the dehydration rate is limited (35% WL, “*Passolata*” type), with the exception of linalool, an increase in content can be seen with certain terpenes, such as geraniol and nerol (data in μg/kg berries), resulting in a terpene profile with a predominance of geraniol.

### 3.3. Glycosylated Volatile Compounds

The glycosylated volatile composition of fresh and dried grapes is shown in Table 4 (µg/100 berries) and Table 5 (µg/kg berries), accounting for 36 compounds detected (27 terpenes, 4 norisoprenoids, and 5 benzenoids). With respect to free volatile compounds reported in Table 2 and Table 3, other compounds have been identified in the enzymatic hydrolysis product of fresh and dehydrated grapes’ glycoconjugate precursors. Relevantly, *trans*-8-hydroxy-nerol, *trans-* and *cis-*8-hydroxy-geraniol, and 2,6-dimethyl-6-hydroxy-2,7-Octadienoic acid are products of H^+^ catalyzed transformation of 8-hydroxy linalool [59,60]. Moreover, compounds belonging to the norisoprenoids class (4-oxo-α-damascone, 3,4-dihydro-3-oxo-actinidol isomer 1, 3-oxo-α-ionol, and vomifoliol) and benzenoids (4-vinylguaiacol, dihydrocoliferyl alcohol, and vanillin) were identified and quantified.

Terpenols (µg/100 berries) were the most abundant class of compounds identified as glycoconjugates in fresh and dehydrated berries. Among these, linalool, 2,6-dimethyl-3,7-octadiene-2,6-diol, geraniol, the two 8-hydroxy linalool isomers, nerol, and *trans*-geranic acid were quantitatively the most important compounds found (Table 4). A slight decrease was generally observed in the concentration of most of the terpene compounds during dehydration. Nevertheless, the difference between the contents of such compounds were significant only for linalool, hydroxy nerol, and hydroxy geraniol (*p <* 0.001) and for geranial, hydroxy citronellol, *p*-menth-1-ene-7,8-diol, *trans*-8-hydroxy geraniol, and *trans*-geranic acid (*p <* 0.05). The content of terpene compounds (from enzymatic hydrolysis of heterosidic fraction) such as linalool and *trans-*geranic acid decreased from the fresh berries to the 65% WL (*p <* 0.05), while geraniol, 3,7-dimethyl-1,7-octadiene-3,6-diol, and *cis*-8-hydroxy-linalool marked non-significant decreases for the same dehydration process (*p >* 0.05). The biggest drop was found in linalool from fresh to 35% WL, but with this decrease, a significant (*p <* 0.05) increase of hydroxy geraniol corresponded and a not significant (*p >* 0.05) increase of 2,6-dimethyl-3,7-octadiene-2,6-diol, *trans*- and *cis*-furan-linalool oxide, hotrienol, and α-terpineol. Nevertheless, at 50% and 65% WL, these compounds also decreased. The evolution of the other terpene compounds was less regular, even if the content of almost all decreased at 65% WL.

In contrast, the content of the norisoprenoids recovered after enzymatic hydrolysis of the heterosidic fraction generally increased from fresh to the dehydrated berries, as well as total benzenoids, although both increases were not significant as total compounds (*p >* 0.05, Table 4).

Considering the concentration in µg/kg berries, the content of total terpenes, norisoprenoids, and benzenoids compounds increased significantly during the dehydration process (Table 5). This significant increase of aroma precursors has been reported elsewhere during both grape ripening [58] and dehydration [4]. Considering the individual glycosylated terpenes, the dehydrated 65% WL berries were significantly richer with respect to the fresh grapes in several terpenes, with values at least tripled in the case of α-terpineol, *cis*-furan-linalool oxide, hydroxy-citronellol, hydroxy-nerol, *trans*-8-hydroxy-linalool, hydroxy-geraniol, *p*-menth-1-ene-7,8-diol, *trans*-8-hydroxy-nerol, and *trans*-8-hydroxy-geraniol.

The percentage variation of the total content of glycosylated terpene compounds (µg/100 berries), compared to fresh grapes during drying, is about −4.2, −16.6, and −28.6% at 35%, 50%, and 65% WL, respectively (Table 4). On the contrary, when data are expressed in µg/kg berries, the variation from fresh grapes is about +45.2%, +71.2%, and 104.3% at each of the three levels of dehydration considered (Table 5).

Glycosylated norisoprenoids and benzenoid compounds significantly increased their content (µg/kg berries) during the dehydration process up to almost four fold at the end of dehydration with respect to their initial concentration in fresh grapes (Table 5). This increase is mainly related to the loss of water associated with the dehydration process, which in turn justifies the higher volatile conjugates content when expressed in μg/kg berries, but a not significant (*p >* 0.05) increase for these compounds’ classes was also found for the data expressed as µg/100 berries.

Generally, glycosylated aromatic compounds are less affected by the dehydration process with respect to the corresponding free aromas, due to the protection of sugar (glucose or disaccharides) against degradation or transformation reactions. Nevertheless, even if to a lesser extent, the decrease in free linalool content (µg/100 berries), as well as in other free terpene compounds, from fresh to dehydrated berries, especially at 35% WL, may be due to free radical oxidation reactions, induced by the presence of ROS, such as hydrogen peroxide, during the dehydration [57]. Besides oxidative reactions, catalysed H^+^ hydrolysis may occur and in the case of glycosylated forms, would first lead to the production of the respective free forms and then to their transformation into H^+^ catalysed forms [57]. In our findings, the low presence of the respective free form transformation products (e.g., hotrienol, α-terpineol, and diols derived from the hydration of linalool, nerol, and geraniol) suggests that oxidation reactions prevailed over H+ catalysed reactions. Considering the compounds’ decrease kinetics, these oxidation reactions reached the maximum rate in the first phase of the dehydration process (35% WL) and then continued more slowly, which may be given by total hydrogen peroxide consumption. A previous work [35] showed that these oxidation reactions can occur independently from the dehydration process, to which the grapes have been subjected. Among the oxidation products, 2,6-dimethyl-3,7-Octadiene-2,6-diol increase was previously reported [61,62]. In our case, its concentration, although found to increase, was not significantly different during dehydration (*p* > 0.05), whereas other oxidation products such as 2,6-dimethyl-7-Octadiene-2,6-diol significantly increased during the dehydration (*p <* 0.05; Table 5). On the other hand, catalysed H^+^ reactions can also be confirmed in the glycosides fraction by the presence of 8-hydroxy nerol and 8-hydroxy geraniol derived from the attack of H^+^ on the -OH group in position 6 of the glycosides of 8-hydroxy linalool [58].

Considering norisoprenoids, their low glycosylated contents are consistent with the characteristics of the aromatic varieties Muscat of Alexandria and Moscato bianco [4,28,29] and their increasing behaviour is barely affected (*p* > 0.05 for all compounds except 3-oxo-α-damascenone) during dehydration when the content of 100 berries is considered. The same trends were found for glycosylated benzenoid compounds. Both classes’ contents were significantly increased by the water loss, leading to a significantly higher concentration when the concentration in µg/kg berries is considered, to a different extent depending on the sampling points. As a consequence, due to the lower degradation suffered by volatile glycosylated forms, their content generally increased from fresh berries to those at different levels of dehydration. Finally, the content of the individual compounds detected in 1 kg of dehydrated grapes was higher than that present in 1 kg of fresh grapes. Therefore, in our experimental conditions, a gain of potential aroma precursors was found. This increase was also important for the glycosylated forms of linalool, the compound most affected by oxidative degradation reactions, underlying the effectiveness of the dehydration process in increasing the aromatic potential besides degradative reactions.

### 3.4. Berry Phenolic Compounds and Mechanical Properties of the Berry Skin

As described in the introduction section, in the production of Sicilian DOC special wines, “*Passolata*”, “*Bionda*”, and “*Malaga*” dehydrated berry grapes are added to a base wine obtained from fresh grapes (< 10% of alcohol) to continue the fermentation process, however in different ratios according to the function of the type and style of wine. Therefore, whole berries are subjected to a maceration process in a medium rich in ethanol, leading to phenolic compounds extraction.

Table 6 summarizes the effect of the dehydration process on total flavonoids content (TFI) in skin, pulp, and seeds and the textural modifications of the skin. Significant differences among fresh grapes and dehydrated grapes at different levels were observed. In particular, when data are expressed as mg/100 berries, an important decrease TFI in grape skins was observed in the first phase (35% WL), followed by an increased concentration. Also, when the data are expressed in mg/kg of berries, a strong loss of flavonoids was detected from fresh and “*Passolata*” dehydrated grapes (35% WL). This decrease (from 308 to 217 mg/kg berries) is mainly to be ascribed to metabolites oxidation, as observed in other studies [5,13,40]. On the contrary, a substantial and significant increase of flavonoid content was observed for 50% WL (“*Bionda*” dehydrated grapes type) and 65% WL (“*Malaga*”), which is related to lower berry weight (i.e., 2.31 g and 1.63 g for 50% and 65% WL, respectively; Table 1). Therefore, the increase in flavonoid compounds can be attributed to the prevalence of the concentration effect on account of water loss over decomposition oxidisation phenomena, although a possible flavonoid biosynthesis cannot be excluded.

However, it is conceivable that part of the flavonoid losses that occurred in the skins are given by some of these compounds passing from the skin to the pulp. The TFI in the berry juice did not report significant (*p* > 0.05) differences from fresh grapes (584 mg/L) to 35% WL and 50% WL (719 and 745 mg/L, respectively), although an increasing trend was evidenced. At 65% WL, the analyses were not performed due to the semisolid and crystalline firmness of the pulp. Higher pulp TFI content is also justified by water loss rather than oxidative degradation; anyway, an extraction from skin to pulp is also given by a decrease in skin hardness. In this sense, break skin force values (Fsk) decreased significantly (*p <* 0.001) during the dehydration process from 0.694 N (fresh grapes) to 0.193 N (dehydrated berries at 65% WL). If on the one hand the reduction in skin hardness led to an acceleration of the dehydrating kinetics of the grapes, as widely demonstrated by scientific literature [13,16], on the other hand, the phenomena of extractability of the phenolic substances are accelerated [63].

During *Passito* winemaking, the dehydrated “*Passolata*”, “*Bionda*”, and “*Malaga*” berries are added as whole berries to a base wine. In this specific context, from a technological point of view, the polyphenols’ contribution given by the seeds can be considered less relevant due to the lower direct contact that they may have with liquid base wine, with respect to a traditional maceration. However, in long macerations, berries degradation may lead to seeds releasing, coming into contact with the base wine (10% *v/v* ethanol), promoting polyphenols extraction. In this sense, the seeds belonging to berries dehydrated at 35% and 50% WL showed similar contents to fresh grapes when flavonoids are expressed as mg/kg berries (about 1200 mg/kg berries). Only at 65% WL did the seeds’ TFI content increase significantly when expressed as concentration on berry weight (> 4000mg/kg berries; 250 mg/100 seeds, *p <* 0.001) due to extreme weight loss sustained by the berries. Therefore, the addition of this dehydration type to the base wine may lead to a higher risk of seeds’ polyphenols extraction, potentially impacting on the chromatic and sensory characteristics [2].

During grape ripening, seeds’ histological and histochemical modifications occurs, with intensive lignification and hardening of the medium integument and the presence of phenolic compounds in the inner integument that can affect phenols release [64]. To our knowledge, histological studies of dehydrated grape seeds that support the evidence of this highest extractability of flavonoids of berries at 65% WL are not still present in literature, although an increase of oligomer and polymer tannin content of seeds during the withering process was already noticed [65].

## 4. Conclusions

Several renowned special wines (i.e., *Icewines*, *Passito*, and *Fortified* wines) are produced worldwide using over-ripe and dehydrated grapes with different technology and winemaking strategies. In Sicily, including Pantelleria island, among the different drying techniques used, a traditional Muscat of Alexandria grape dehydration process is carried out, leading to grapes with high weight loss (from 35% up to 65% weight loss), also with the aid of sodium hydroxide grape soaking. This preliminary treatment, together with the characteristic environmental conditions, permits a very fast dehydration process. In our experimental conditions, the alkaline pre-treatment conducted prior to dehydration simulating typical thermo-hygrometic conditions, allowed us to reach up to 65% weight loss in just 13 days. However, this study highlighted how this process strongly impacted the chemical-physical grape characteristics. In particular, base, aromatic, and phenolic parameters were analysed from both physiological (data in µg/100 berries) and technological (data in µg/kg berries) points of view. Under the physiological point of view, a degradation of many aromatic compounds occurred to different extents depending on the sampling point, and the modifications were related to the susceptibility of each compound to oxidation and hydrolysis-related reactions. In contrast, from a technological point of view, an increase of many compounds was observed due to the concentration effect (i.e., berry dehydration). Moreover, the rate of dehydration had a strong impact on the volatile composition and profile of Muscat of Alexandria grapes, leading to different possible final products. In fact, volatile markers such as linalool, geraniol, and hotrienol were present above their perception threshold, but in a different ratio depending on the weight loss reached. As expected, free aromas were more prone to degradation, whereas glycosylated forms were less reduced and they were concentrated on grape weight basis depending on the water loss, leading to an increased grape aromatic potential.

Since whole berries are used in *Passito* production, the phenolic composition was also investigated: the concentration changed during the dehydration process and the three different weight loss levels considered showed different flavonoid content. In particular, “*Malaga*” (65% weight loss) showed a consistently higher phenolic concentration (on berry weight) together with decreased skin hardness, which may be taken in consideration during winemaking to avoid increased astringency or color hue. Generally, part of phenols is lost by oxidation and, to a lesser extent, is transferred from skin to pulp because of change in tissue characteristics, which determine a decrease in the skin hardness values.

This study permitted us to increase the knowledge about the effects of this peculiar dehydration process applied to *cv* Muscat of Alexandria grape in the production of several DOC wines produced on Sicily island.

## Figures and Tables

**Table 1 foods-09-00666-t001:** Evolution of Muscat Alexandria grapes composition during dehydration.

Days		0	5	8	13	Sign
Weight Loss%		Fresh Grapes	35%	50%	65%	
Mean berry weight (g)		4.97 ± 0.54d	3.09 ± 0.12c	2.31 ± 0.19b	1.63 ± 0.11a	***
	Δ*%*		−37.9	−53.6	−67.2	

Reducing sugars (g/L)		189 ± 8a	266 ± 6b	344 ± 8c	387 ± 10d	***
	Δ*%*		40.7	82.0	104.8	

Total acidity (g/L as tartaric acid)		5.40 ± 0.16b	3.97 ± 0.16a	5.15 ± 0.08b	5.25 ± 0.21b	***
	Δ*%*		−26.5	−4.6	−2.8	

pH		3.20 ± 0.04	3.26 ± 0.02	3.25 ± 0.01	3.24 ± 0.02	ns
	Δ*%*		1.9	1.6	1.2	

Data are expressed as average ± standard deviation (*n* = 3). Different Latin letters within the same row indicate significant differences (*p* < 0.001, ***) among different dehydration levels (Tukey-b test). ns: not significant. Δ% compared to fresh grapes. pH and total acidity data are reported from Corona et al. [7].

**Table 2 foods-09-00666-t002:** Evolution of free volatile compounds of Muscat Alexandria grapes during dehydration (μg/100 berries).

Days	0	5	8	13	Sign
Weight Loss%	Fresh Grapes	35%	50%	65%	
**Terpenes compounds**					
trans-Furan-linalool oxide	15.4 ± 2.3	19.7 ± 7.4	23.5 ± 6.9	18.2 ± 3.9	ns
cis-Furan-linalool oxide	34.9 ± 8.7	29.7 ± 9.0	34.1 ± 6.7	18.9 ± 4.6	ns
Linalool	531.7 ± 32.0b	52.6 ± 10.7a	42.5 ± 5.3a	50.8 ± 1.9a	***
Hotrienol	28.5 ± 3.9	22.8 ± 9.1	39.6 ± 7.5	39.1 ± 11.4	ns
α-Terpineol	20.2 ± 3.4a	26.9 ± 2.5b	32.5 ± 5.2c	37.3 ± 2.5d	***
trans-Piran-linalool oxide	192.3 ± 12.2d	99.8 ± 9.8c	43.7 ± 12.9b	24.5 ± 8.5a	***
cis-Piran-linalool oxide	94.0 ± 12.4b	116.2 ± 23.7b	107.1 ± 13.5b	48.7 ± 9.8a	***
Citronellol	3.3 ± 0.9	1.7 ± 0.6	2.0 ± 0.6	2.2 ± 0.1	ns
Nerol	36.1 ± 3.4b	39.9 ± 2.1b	11.1 ± 4.5a	8.4 ± 2.9a	***
Geraniol	268.9 ± 31.4b	214.6 ± 27.9b	79.6 ± 31.8a	28.1 ± 12.4a	***
2,6-dimethyl-3,7-Octadiene-2,6-diol	390.5 ± 67.0	554.3 ± 205.8	522.5 ± 154.3	555.5 ± 96.2	ns
2,6-dimethyl-7-Octadiene-2,6-diol	5.2 ± 1.7a	10.2 ± 4.7a	16.3 ± 4.5ab	26.0 ± 6.5b	***
3,7-dimethyl-1,7-Octadiene-3,6-diol	94.0 ± 12.3c	66.3 ± 8.7bc	39.5 ± 21.9ab	26.6 ± 7.7a	***
trans-8-hydroxy-linalool	32.1 ± 5.8b	15.4 ± 5.9a	17.4 ± 4.9a	23.4 ± 6.9ab	*
cis-8-hydroxy-linalool	17.8 ± 4.6	19.8 ± 7.6	8.9 ± 1.9	8.2 ± 5.7	ns
hydroxy-Geraniol	24.4 ± 11.2	16.2 ± 4.6	9.1 ± 1.7	9.9 ± 6.0	ns
trans-Geranic acid	216.6 ± 49.6c	137.2 ± 22.9b	31.5 ± 15.2a	18.8 ± 17.5a	***
**∑ Terpenes**	2005.9 ± 262.7b	1443.4 ± 362.8ab	1061.0 ± 299.1a	944.4 ± 204.5a	**
**Δ%**		−28.0	−47.1	−52.9	
**Benzenoids**					
2-Phenylethanol	79.5 ± 14.3a	115.2 ± 50.0ab	60.8 ± 10.4a	162.7 ± 26.0b	*
Benzyl Alcohol	8.6 ± 3.5ab	10.6 ± 2.6b	2.5 ± 1.7a	4.7 ± 2.5ab	*
**∑ Benzenoids**	88.1 ± 17.8a	125.8 ± 52.7ab	63.4 ± 12.1a	167.5 ± 28.4b	*
**Δ%**		42.7	−28.1	90.0	

Data are expressed as average ± standard deviation (*n* = 3). WL% is the weight loss percentage. Different Latin letters within the same row indicate significant differences (*p* < 0.05, 0.01, and 0.001, respectively *, **, ***) among different dehydration levels (Tukey-b test). Δ% compared to fresh grapes.

**Table 3 foods-09-00666-t003:** Evolution of free volatile compounds of Muscat Alexandria grapes during dehydration (μg/kg berries).

Days	0	5	8	13	Sign
Weight Loss%	Fresh Grapes	35%	50%	65%	
**Terpenes**					
*trans*-Furan-linalool oxide	31.1 ± 5.4a	64.4 ± 26.9ab	101.1 ± 22.7b	111.0 ± 16.9b	***
*cis*-Furan-linalool oxide	70.2 ± 16.6a	96.9 ± 33.5ab	148.9 ± 35.1ab	115.0 ± 21.6b	*
Linalool	1074.1 ± 71.2c	170.8 ± 36.7a	186.0 ± 37.3a	312.2 ± 16.6b	***
Hotrienol	57.5 ± 8.1a	73.9 ± 29.6a	174.3 ± 47.3b	237.2 ± 55.7b	***
α-Terpineol	41.3 ± 10.4a	87.5 ± 11.5a	142.7 ± 34.3b	229.1 ± 17.4c	***
*trans*-Piran-linalool oxide	388.3 ± 22.3b	323.5 ± 32.3b	192.5 ± 71.4a	149.2 ± 45.0a	***
*cis*-Piran-linalool oxide	189.6 ± 21.9a	378.5 ± 92.4b	469.3 ± 94.7b	296.5 ± 40.5ab	***
Citronellol	6.6 ± 2.0a	5.5 ± 1.7a	9.0 ± 3.1a	13.7 ± 0.2b	***
Nerol	72.7 ± 1.1a	129.6 ± 11.9b	47.3 ± 15.3a	50.6 ± 14.5a	***
Geraniol	541.1 ± 28.6b	694.9 ± 85.7b	339.5 ± 109.2a	169.5 ± 64.5a	***
2,6-dimethyl-3,7-Octadiene-2,6-diol	784.7 ± 93.6a	1813.1 ± 747.7ab	2308.8 ± 833.4bc	3390.4 ± 371.8c	***
2,6-dimethyl-7-Octadiene-2,6-diol	10.3 ± 2.1a	33.5 ± 16.6ab	72.1 ± 25.7b	158.6 ± 32.6c	***
3,7-dimethyl-1,7-Octadiene-3,6-diol	191.2 ± 37.1	215.6 ± 36.0	177.2 ± 112.4	161.6 ± 37.7	ns
*trans*-8-hydroxy-linalool	64.4 ± 7.2a	50.4 ± 21.3a	77.0 ± 27.7a	142.1 ± 33.2b	***
*cis*-8-hydroxy-linalool	35.4 ± 6.7	64.9 ± 27.6	38.8 ± 8.4	49.0 ± 31.5	ns
hydroxy-Geraniol	48.1 ± 16.5	52.7 ± 17.3	39.7 ± 9.4	59.2 ± 33.0	ns
*trans*-Geranic acid	432.8 ± 56.2b	443.8 ± 66.7b	134.2 ± 54.9a	112.0 ± 98.6a	***
**∑ Terpenes**	4039.2 ± 407.0a	4699.6 ± 1295.4ab	4658.2 ± 1542.4ab	5756.8 ± 931.2b	*
**Δ%**		18.0	25.4	56.5	
**Alcohols**					
2-Phenylethanol	161.5 ± 38.0a	371.2 ± 156.1a	267.4 ± 67.4a	1001.0 ± 170.3b	***
Benzyl Alcohol	17.1 ± 5.9	34.3 ± 9.2	11.0 ± 7.2	28.5 ± 13.1	ns
**∑ Alcohols**	178.6 ± 44.0a	405.5 ± 165.3a	278.4 ± 74.6a	1029.5 ± 183.5b	***
**Δ%**		127.0	55.8	476.3	

Data are expressed as average ± standard deviation (*n* = 3). WL% is the weight loss percentage. Different Latin letters within the same row indicate significant differences (*p* < 0.05 and 0.001, respectively *, ***) among different dehydration levels (Tukey-b test). ns: not significant. Δ% compared to fresh grapes.

**Table 4 foods-09-00666-t004:** Evolution of glycosylated volatile compounds of Muscat Alexandria grapes during dehydration (µg/100 berries).

Days	0	5	8	13	Sign
Weight Loss%	Fresh Grapes	35%	50%	65%	
**Terpenes Compounds**					
trans-Furan-linalool oxide	161.5 ± 13.3	254.9 ± 99.8	214.9 ± 66.8	170.1 ± 69.1	ns
cis-Furan-linalool oxide	36.2 ± 6.8	36.0 ± 6.6	36.8 ± 9.2	34.2 ± 7.5	ns
Linalool	976.4 ± 90.0b	601.8 ± 73.2a	415.6 ± 109.2a	381.7 ± 92.4a	***
Hotrienol	43.2 ± 7.8	61.7 ± 30.3	43.3 ± 3.9	22.2 ± 4.3	ns
Neral	7.4 ± 0.4	10.2 ± 3.5	8.5 ± 3.0	5.2 ± 0.9	ns
α-Terpineol	51.6 ± 6.0	84.9 ± 21.6	69.4 ± 15.4	63.5 ± 7.0	ns
Geranial	18.7 ± 3.9b	16.8 ± 2.4b	14.2 ± 1.4ab	10.6 ± 1.4a	*
trans-Piran-linalool oxide	58.4 ± 7.5	51.4 ± 18.7	37.6 ± 9.4	26.3 ± 10.0	ns
cis-Piran-linalool oxide	16.4 ± 3.7	26.1 ± 9.1	29.7 ± 10.6	18.5 ± 10.1	ns
Citronellol	14.3 ± 0.2	12.0 ± 0.5	11.9 ± 7.1	8.3 ± 0.3	ns
Nerol	227.8 ± 100.3	233.0 ± 31.0	183.4 ± 23.8	167.6 ± 13.6	ns
Geraniol	771.4 ± 279.7	620.7 ± 83.9	495.6 ± 48.8	423.0 ± 36.6	ns
2,6-dimethyl-3,7-Octadiene-2,6-diol	784.9 ± 174.3	933.0 ± 188.8	884.4 ± 217.2	664.9 ± 150.0	ns
2,6-dimethyl-7-Octadiene-2,6-diol	121.7 ± 114.0	96.7 ± 16.4	108.3 ± 26.5	112.0 ± 16.4	ns
3,7-dimethyl-1,7-Octadiene-3,6-diol	92.6 ± 2.1	73.6 ± 19.6	72.3 ± 35.4	52.7 ± 12.2	ns
hydroxy Citronellol	5.0 ± 3.3a	11.9 ± 1.9ab	12.1 ± 2.9ab	15.2 ± 2.6b	*
8-hydroxy dihydrolinalool	67.5 ± 12.7	64.5 ± 8.7	65.7 ± 16.6	60.5 ± 9.9	ns
hydroxy Nerol	4.9 ± 3.4a	17.5 ± 3.6b	17.8 ± 1.9b	23.0 ± 4.8b	***
trans-8-hydroxy-linalool	257.5 ± 53.8	256.9 ± 56.7	255.9 ± 66.1	241.4 ± 47.6	ns
cis-8-hydroxy-linalool	194.3 ± 70.1	151.5 ± 111.0	145.8 ± 11.6	107.5 ± 20.5	ns
hydroxy Geraniol	75.9 ± 48.0a	194.3 ± 20.2c	122.5 ± 22.3ab	181.8 ± 13.9bc	***
trans-Geranic acid	433.3 ± 41.9b	395.5 ± 72.0ab	395.0 ± 22.4ab	288.7 ± 26.0a	*
p-menth-1-ene-7,8-diol	n.d.	14.3 ± 3.9ab	9.5 ± 8.8a	22.6 ± 4.3b	*
trans-8-hydroxy-Nerol	11.4 ± 2.9	20.1 ± 6.1	22.8 ± 5.2	22.6 ± 4.7	ns
cis-8-hydroxy-Geraniol	81.4 ± 27.8	54.4 ± 6.0	57.1 ± 5.4	56.1 ± 10.6	ns
trans-8-hydroxy-Geraniol	28.7 ± 8.2a	43.8 ± 8.5ab	47.8 ± 4.5ab	52.9 ± 7.5b	*
2,6-dimethyl-6-hydroxy-2,7-octadienoic acid	23.1 ± 9.2	34.9 ± 22.3	28.4 ± 9.5	26.3 ± 1.7	ns
**∑ Terpenes**	4565.3 ± 1091.2	4372.5 ± 926.1	3806.6 ± 764.6	3259.2 ± 585.7	ns
**Δ%**		−4.2	−16.6	−28.6	
**Norisoprenoids**					
4-oxo-α-Damascenone	9.1 ± 0.1a	10.3 ± 4.3ab	14.7 ± 3.4ab	20.1 ± 4.3b	*
3,4-dihydro-3-oxo-actinidol 1	6.6 ± 2.7	13.2 ± 2.1	10.5 ± 2.1	10.3 ± 1.4	ns
3-oxo-α-ionol	60.0 ± 0.4	53.7 ± 6.1	57.2 ± 8.3	52.7 ± 7.9	ns
Vomifoliol	82.0 ± 10.4	120.9 ± 47.8	118.5 ± 40.4	153.2 ± 36.5	ns
**∑ Norisoprenoids**	157.8 ± 13.7	198.0 ± 60.3	200.9 ± 54.1	236.4 ± 50.1	ns
**Δ%**		25.5	27.3	49.8	
**Benzenoids**					
Benzyl Alcohol	27.5 ± 17.0	39.7 ± 14.5	26.3 ± 16.6	48.7 ± 11.6	ns
2-Phenylethanol	57.7 ± 6.4a	83.9 ± 21.2a	64.7 ± 28.4a	148.4 ± 50.1b	*
4-Vinylguaiacol	115.6 ± 9.2	172.6 ± 85.3	150.8 ± 25.6	163.4 ± 23.4	ns
Vanillin	12.6 ± 0.1	16.0 ± 2.4	10.4 ± 2.0	15.1 ± 2.5	ns
dihydrocoliferyl alcohol	44.3 ± 7.2	65.2 ± 23.4	72.0 ± 8.2	45.1 ± 9.7	ns
**∑ Benzenoids**	257.7 ± 39.9	377.4 ± 146.9	323.2 ± 80.8	420.7 ± 97.4	ns
**Δ%**		46.5	25.8	63.3	

Data are expressed as average ± standard deviation (*n* = 3). WL% is the weight loss percentage. Different Latin letters within the same row indicate significant differences (*p* < 0.05 and 0.001, respectively *, ***) among different dehydration levels (Tukey-b test). ns: not significant. Δ% compared to fresh grapes.

**Table 5 foods-09-00666-t005:** Evolution of glycosylated volatile compounds of Muscat Alexandria grapes during dehydration (µg/kg berries).

Days	0	5	8	13	Sign
Weight Loss%	Fresh Grapes	35%	50%	65%	
**Terpenes Compounds**					
trans-Furan-linalool oxide	346.8 ± 40.7	817.8 ± 296.4	944.6 ± 352.1	1029.1 ± 381.8	ns
cis-Furan-linalool oxide	77.4 ± 11.8a	116.5 ± 20.0ab	162.6 ± 53.5ab	208.5 ± 32.7b	*
Linalool	2090.9 ± 119.9	1956.7 ± 314.8	1827.0 ± 564.9	2322.5 ± 441.6	ns
Hotrienol	93.0 ± 20.0	197.8 ± 90.6	189.1 ± 31.0	137.0 ± 31.3	ns
Neral	15.8 ± 0.3	32.7 ± 10.3	36.9 ± 13.6	31.7 ± 4.1	ns
α-Terpineol	110.5 ± 9.0a	273.4 ± 60.0b	305.5 ± 92.6b	388.7 ± 18.1b	***
Geranial	39.9 ± 7.0a	54.4 ± 5.7ab	62.0 ± 9.7b	65.1 ± 6.4b	*
trans-Piran-linalool oxide	125.5 ± 20.3	165.0 ± 55.0	164.7 ± 49.4	159.1 ± 53.6	ns
cis-Piran-linalool oxide	35.0 ± 6.6	84.0 ± 26.5	130.7 ± 53.7	111.5 ± 57.9	ns
Citronellol	30.7 ± 0.6	38.7 ± 2.0	53.5 ± 34.3	50.7 ± 1.5	ns
Nerol	484.8 ± 198.1a	754.1 ± 90.4ab	804.4 ± 170.5ab	1026.7 ± 18.9b	*
Geraniol	1644.1 ± 542.1	2014.8 ± 321.1	2159.2 ± 281.3	2590.8 ± 66.8	ns
2,6-dimethyl-3,7-Octadiene-2,6-diol	1677.0 ± 315.1	3020.9 ± 585.5	3895.2 ± 1270.6	4066.6 ± 817.1	ns
2,6-dimethyl-7-Octadiene-2,6-diol	256.8 ± 235.3a	313.4 ± 52.6a	476.8 ± 154.0ab	685.5 ± 79.8b	*
3,7-dimethyl-1,7-Octadiene-3,6-diol	198.5 ± 2.5	237.3 ± 55.4	322.0 ± 183.5	322.1 ± 63.9	ns
hydroxy Citronellol	10.7 ± 6.6a	38.5 ± 4.7b	52.3 ± 11.2b	93.5 ± 15.8c	***
8-hydroxy dihydrolinalool	144.2 ± 22.1a	208.8 ± 24.2ab	289.9 ± 97.2ab	370.8 ± 54.7b	*
hydroxy Nerol	10.4 ± 6.9a	56.4 ± 9.6b	78.0 ± 13.7b	141.1 ± 28.0c	***
trans-8-hydroxy-linalool	550.4 ± 96.2a	834.9 ± 200.8ab	1129.3 ± 386.1ab	1474.7 ± 237.9b	*
cis-8-hydroxy-linalool	414.2 ± 135.8	499.7 ± 385.5	637.0 ± 99.9	661.1 ± 128.7	ns
hydroxy Geraniol	160.9 ± 97.3a	629.7 ± 67.8b	528.6 ± 59.1b	1115.5 ± 64.5c	***
trans-Geranic acid	927.7 ± 57.5a	1281.0 ± 226.2ab	1725.3 ± 240.2b	1770.6 ± 116.9b	***
p-menth-1-ene-7,8-diol	n.d.	46.5 ± 14.8a	42.2 ± 40.9a	138.8 ± 25.9b	***
trans-8-hydroxy-Nerol	24.4 ± 5.4a	64.7 ± 17.9ab	100.4 ± 30.2bc	138.7 ± 28.9c	***
cis-8-hydroxy-Geraniol	175.7 ± 65.8a	176.5 ± 21.7a	249.9 ± 41.9ab	343.2 ± 54.9b	*
trans-8-hydroxy-Geraniol	61.3 ± 15.4a	142.3 ± 30.1b	207.9 ± 19.7b	324.2 ± 39.9c	***
2,6-dimethyl-6-OH-2,7-Octadienoic acid	49.2 ± 17.9	114.6 ± 77.2	125.8 ± 52.5	161.3 ± 8.3	ns
**∑ Terpenes**	9755.4 ± 2056.1a	14171.0 ± 3057.7ab	16700.8 ± 4407.3ab	19928.6 ± 2580.0b	**
**Δ%**		45.3	71.2	104.3	
**Norisoprenoids**					
4-oxo-α-Damascenone	19.5 ± 0.4a	33.6 ± 15.5a	64.5 ± 17.3a	122.7 ± 20.8b	***
3,4-dihydro-3-oxo-actinidol 1	14.1 ± 5.4a	42.9 ± 8.3b	45.4 ± 8.0b	63.4 ± 9.1b	***
3-oxo-α-ionol	128.7 ± 3.6a	173.7 ± 16.7ab	249.6 ± 46.6bc	323.1 ± 41.4c	***
Vomifoliol	175.5 ± 16.2a	394.7 ± 168.8a	524.3 ± 221.5ab	943.3 ± 231.9b	*
**∑ Norisoprenoids**	337.9 ± 25.5a	644.9 ± 209.4a	883.8 ± 293.4ab	1452.4 ± 303.2b	***
**Δ%**		90.9	161.6	329.8	
**Benzenoids**					
2-Phenylethanol	123.9 ± 18.0a	272.7 ± 74.9a	288.3 ± 146.6a	919.8 ± 331.8b	***
Benzyl Alcohol	58.4 ± 34.4a	129.7 ± 52.6ab	118.5 ± 83.1ab	302.4 ± 90.1b	*
4-Vinylguaiacol	248.3 ± 28.4a	562.7 ± 288.4ab	657.7 ± 133.5ab	1009.3 ± 191.2b	*
Vanillin	26.9 ± 0.7a	51.9 ± 7.3b	45.8 ± 12.7b	92.3 ± 9.3b	***
dihydrocoliferyl alcohol	94.8 ± 12.2a	212.7 ± 83.9ab	311.5 ± 10.9b	279.9 ± 81.0b	*
**∑ Benzenoids**	552.3 ± 93.6a	1229.7 ± 507.1ab	1421.8 ± 386.7ab	2603.7 ± 703.4b	**
**Δ%**		122.7	157.4	371.4	

Data are expressed as average ± standard deviation (*n* = 3). WL% is the weight loss percentage. Different Latin letters within the same row indicate significant differences (*p* < 0.05, 0.01, and 0.001, respectively *, **, ***) among different dehydration levels (Tukey-b test). ns: not significant. Δ% compared to fresh grapes.

**Table 6 foods-09-00666-t006:** Evolution of skin hardness and polyphenols in the berry skin, seeds, and juice obtained from the pulp of Muscat of Alexandria grapes during dehydration.

Days		0	5	8	13	Sign
Weight Loss%		Fresh Grapes	35%	50%	65%	
**Berry skin**						
Total flavonoids index		153 ± 18c	71 ± 3a	93 ± 1ab	101 ± 10b	***
(mg/100 berries as (+)-catechin)	Δ%		−53.4	−39.4	−34.0	

Total flavonoids index		308 ± 41b	217 ± 8a	332 ± 19b	544 ± 33c	***
(mg/kg berries as (+)-catechin)	Δ%		-29.6	+7.9	+76.9	

Fsk (N)		0.694 ± 0.138c	0.572 ± 0.345bc	0.469 ± 0.337b	0.193 ± 0.198a	***
	Δ%		−17.6	−32.4	−72.2	
**Berry seeds**						
Total flavonoids index		273 ± 62b	122 ± 26a	130 ± 22a	249 ± 22b	**
(mg/100 seeds as (+)-catechin)	Δ%		−55.3	−52.4	−8.9	

Total flavonoids index		1074 ± 402a	1135 ± 235a	1403 ± 197a	4040 ± 547b	***
(mg/kg berries as (+)-catechin)	Δ%		+5.6	+30.6	+276.0	
**Berry juice**						
Total flavonoids index		584 ± 73	719 ± 83	745 ± 98	− ^#^	ns
(mg/L as (+)-catechin)	Δ%		+23.1	+27.5	− ^#^	

Data are expressed as average ± standard deviation (*n* = 3 for total flavonoids, *n* = 30 for Fsk and Wsk). Different Latin letters within the same row indicate significant differences (*p* < 0.01 and 0.001, respectively **, ***) among different dehydration levels (Tukey-b test). ns: not significant. Δ% compared to fresh grapes. ^#^ The grape condition at 65%WL did not allow the evaluation of juice total flavonoids.

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
