# Peer review of "Influence of Different Dehydration Levels on Volatile Profiles, Phenolic Contents and Skin Hardness of Alkaline Pre-Treated Grapes cv Muscat of Alexandria (Vitis vinifera L.)"

_foods, 2020, doi:10.3390/foods9050666_

Round 1

Reviewer 1 Report

In my opinion, under tables 1-6, instead of a word "column", there should be a "row" in the sentence: Different Latin letters within the same column indicate significant …

Author Response

Thank you for your positive comments. Your comments have been considered and changes in the text is reported using the "Track Changes" function in Microsoft Word (P 12; P 14; P 17; P 20; P 22; P 26)

Reviewer 2 Report

The authors present interesting results that some questions should be addressed:

  1. There seems to be no particular benefit to show the results on a 100 berries base, we advise to reduce paper length and increase clarity by showing only results on a por gram basis. In cases where authors think it is relevant, it may be done on the text without a whole table.
  2. In the abstract explain what is meant by "Aroma compounds degradation ...concentration effect."
  3. Why there are no data on total or fixed acidity? The relatively low pH (about 3.20) indicates a high titrable acidity and it would be relevant for the readers to understand the acid/ sugar balance.

Other minor questions:

  1. Replace Sicilia by Sicily
  2. Table footnotes appear sometimes without clear separation from the text.
  3. Sfursat is not explained in the text, although familiar to an italian audience, should be replaced by a more known denomination or deleted.

Author Response

  1. Thank you for your positive comments. All your comments have been considered and changes in the text are reported using the "Track Changes" function in Microsoft Word.

Comments and Suggestions for Authors

The authors present interesting results that some questions should be addressed:

There seems to be no particular benefit to show the results on a 100 berries base, we advise to reduce paper length and increase clarity by showing only results on a por gram basis. In cases where authors think it is relevant, it may be done on the text without a whole table.

  1. With, respect, we believe it is essential to leave comments and data tables in mg/100 berries. This data are important to evaluate the evolution of volatile compounds from a physiological point of view (biosynthesis/degradation ratio).

In the abstract explain what is meant by "Aroma compounds degradation ...concentration effect."

  1. This sentence has been deleted (P 1)

Why there are no data on total or fixed acidity? The relatively low pH (about 3.20) indicates a high titrable acidity and it would be relevant for the readers to understand the acid/ sugar balance.

  1. The M&M (P 7) and data of “total acidity” are added in table 1 (P 11) and the comments are added in the text (P 13). Thank you for the suggenstion!

Other minor questions:

Replace Sicilia by Sicily

  1. Changed as suggested (P1; P 6; P 7; 242, P 29;). Thank you for the suggenstion!

Table footnotes appear sometimes without clear separation from the text.

  1. Changed as suggested

Sfursat is not explained in the text, although familiar to an italian audience, should be replaced by a more known denomination or deleted.

  1. In order to ovoid misunderstanding we have deleted ‘Sfursat’ term (L 221, P 27).

Reviewer 3 Report

This paper deals with the influence of different dehydration levels on volatile profiles, phenolic contents and skin hardness of alkaline pre-treated grapes cv Muscat of Alexandria (Vitis vinifera L.). The experiments have been very well designed and carried and the methodologies used in this study are specific and the experimental is very good explained. The topic is interesting and important for the wine industry, to know the influence of the dehydration levels on the grape quality for wine production, namely volatile and phenolic composition.

Table 2, Table 3, Table 4 and Table 5: “trans and cis.” should be rewritten in italic.

Author Response

Comments and Suggestions for Authors

This paper deals with the influence of different dehydration levels on volatile profiles, phenolic contents and skin hardness of alkaline pre-treated grapes cv Muscat of Alexandria (Vitis vinifera L.). The experiments have been very well designed and carried and the methodologies used in this study are specific and the experimental is very good explained. The topic is interesting and important for the wine industry, to know the influence of the dehydration levels on the grape quality for wine production, namely volatile and phenolic composition.

Table 2, Table 3, Table 4 and Table 5: “trans and cis.” should be rewritten in italic.

  1. Thank you for your positive comments. Your comments have been considered and in the text is reported using the "Track Changes" function in Microsoft Word (P 13-14; P 17; P 19-20; P 21-22)
